# Bedtime Monitoring for Fall Detection and Prevention in Older Adults

**DOI:** 10.3390/ijerph19127139

**Published:** 2022-06-10

**Authors:** Jesús Fernández-Bermejo Ruiz, Javier Dorado Chaparro, Maria José Santofimia Romero, Félix Jesús Villanueva Molina, Xavier del Toro García, Cristina Bolaños Peño, Henry Llumiguano Solano, Sara Colantonio, Francisco Flórez-Revuelta, Juan Carlos López

**Affiliations:** 1Computer Architecture and Networks Group, University of Castilla-La Mancha, Paseo de la Universidad, 4, 13005 Ciudad Real, Spain; javier.dorado@uclm.es (J.D.C.); mariajose.santofimia@uclm.es (M.J.S.R.); felix.villanueva@uclm.es (F.J.V.M.); xavier.deltoro@uclm.es (X.d.T.G.); cristina.bolanos@uclm.es (C.B.P.); henry.llumiguano@uclm.es (H.L.S.); juancarlos.lopez@uclm.es (J.C.L.); 2Institute of Information Science and Technologies, National Research Council of Italy, Via G. Moruzzi, 1, 56124 Pisa, Italy; sara.colantonio@isti.cnr.it; 3Department of Computing Technology, University of Alicante, 03080 Alicante, Spain; francisco.florez@ua.es

**Keywords:** fall detection, fall prevention, wearable sensors, bedtime monitoring, assisted living

## Abstract

Life expectancy has increased, so the number of people in need of intensive care and attention is also growing. Falls are a major problem for older adult health, mainly because of the consequences they entail. Falls are indeed the second leading cause of unintentional death in the world. The impact on privacy, the cost, low performance, or the need to wear uncomfortable devices are the main causes for the lack of widespread solutions for fall detection and prevention. This work present a solution focused on bedtime that addresses all these causes. Bed exit is one of the most critical moments, especially when the person suffers from a cognitive impairment or has mobility problems. For this reason, this work proposes a system that monitors the position in bed in order to identify risk situations as soon as possible. This system is also combined with an automatic fall detection system. Both systems work together, in real time, offering a comprehensive solution to automatic fall detection and prevention, which is low cost and guarantees user privacy. The proposed system was experimentally validated with young adults. Results show that falls can be detected, in real time, with an accuracy of 93.51%, sensitivity of 92.04% and specificity of 95.45%. Furthermore, risk situations, such as transiting from lying on the bed to sitting on the bed side, are recognized with a 96.60% accuracy, and those where the user exits the bed are recognized with a 100% accuracy.

## 1. Introduction

Life expectancy has been increasing over the years [1], mainly due to various scientific and technological achievements. This fact poses a series of challenges that can be addressed from scientific and technological development, improving the lives of the aging population, either by preventive techniques that encourage active aging, or by monitoring methods that inform about the state of health at all times, in order to provide them with the necessary care.

According to [2], older adults may develop various health problems, including somatic diseases and chronic conditions, physical function problems, psychological and cognitive problems, and even various social difficulties. Among these conditions, physical difficulties, such as motor impairment, and difficulty in performing activities of daily living (ADL) are very widespread, with more than 70% of older adults (>85 years) suffering such problems. These mobility impairments, along with other problems that may appear with age, such as dementia, become factors that might increase the likelihood of falling and the consequences that this may entail.

Falls in Spain, as in the vast majority of countries, are a public health problem affecting the older adult population. According to [3,4], it can be estimated that in Spain, 28.4% of older adults over 65 suffer one or more falls per year, and 9.9% of these people have two or more falls per year. It is worth mentioning that those most prone to fall are people with a disability or those who live alone. Injuries and superficial contusions, fractures and, in very extreme, cases of craniocephalic traumatism can be observed as the main consequences after suffering a fall. Moreover, falls can also produce psychological sequelae, such as the fear of falling again, also known as the fear of falling syndrome [3].

Despite the advances in technology and communications, still there is not a commercial solution that successfully addresses the problem of falls in older adults. The balance between provided or perceived functionality and privacy issues, as well as direct and indirect costs, are major barriers for the success of technological solutions for fall detection and prevention [5].

The aim of this work is to present a system that can overcome these limitations, offering a fall detection and fall prevention system that can satisfy the user’s requirements. This system is privacy aware and runs in real time in an unobtrusive way with adequate performance.

## 2. Related Work

The analysis of the state of the art for fall detection and prevention systems brings into light the use of three different type of hardware solutions: those based on wearable sensors, those based on ambient sensors and the ones based on computer vision systems [6]:Wearable sensors: This type of solution is based on sensors placed on the user’s body. IMUs, consisting of accelerometers and gyroscopes that measure the user’s movements, are the most common ones. Within this type of sensor, smartphones have gained special relevance, as they have a powerful processing unit and at the same time they have accelerometers and gyroscopes.Ambient sensors: Sensors of this type are usually stationary, being located in a specific location of the user’s house. These types of sensors can be very different, ranging from pressure sensors, vibration sensors, presence sensors, etc. The main limitation of such sensors is that they are easily affected by external environmental factors, thus producing false alarms.Vision-based systems: These systems employ a variety of computer vision techniques to detect person fall or to perform gait analysis to determine the risk of falling of a certain individual. Vision-based systems can make use of different types of cameras, ranging from a single RGB camera, multiple cameras and depth cameras, such as a Kinect. However, the major problem faced by these solutions is the lack of privacy for the user.

More specifically, the works using wearable sensors can be organized into three main subcategories: those that use bracelets or bands, either for the wrist, the waist or the ankle; those that embed the sensor into some kind of smart clothing; and those that rely on smart phones. Table 1 summarizes in which category falls in each of the works reviewed here. The work in [7] presents a device based on a wristband that combines a threshold-based method with machine learning approaches. This work also considers user acceptance as a factor for the success of the proposed solution. The use of a wristband is appointed as being an asset because the fall detection hardware can go unnoticed. The solution in [8] proposes the use of an accelerometer sensor located in the user ankle, capable of identifying three different states—fall, falling risk and activity of daily living—using a recurrent neural network algorithm. In the same line, the use of the threshold-based method is proposed in [9] but rather than in a wristband, it resorts to a footwear-based device. Although no user acceptance is evaluated in this work, the requirement of being as unobtrusive as possible led the design of this solution. Similarly, the work in [10] combines electromyography and plantar pressure signals. Four surface electromyography (sEMG) sensors are employed to capture the signals of four body muscles. Then, a machine learning classifier is employed to detect movements as gait or falls. The work in [11] also tracks the foot movements using a inertial measurement unit (IMU) with 3D accelerometers, gyroscope and magnetometers. The information collected by this sensor is intended to categorize a falling person or a not falling person. The solution in [12] is more versatile, as it does not require the sensor to be placed in a specific location. On the contrary, the proposed IMU sensor can be located in any part of the body. Although the obtained accuracy for fall detection is promising, the system is not yet ready to work in real time. The work in [13] uses the sensors provided by an OPAL sensor to collect the acceleration generated by the body movement and, from that information and using a machine learning classifier, detect a fall. The work in [14] presents a solution based on smart clothing that collects gravitational acceleration to detect a fall using a hidden Markov model [15]. This work differs among four different states: balanced, imbalanced, falling, and normal state. This study also analyzes user acceptance from the point of view of end users but also caregivers, yielding high rates of acceptance. The work in [16] proposes the use of the accelerometer sensor to collect acceleration data that are is first processed by a threshold-based method and then, those over the threshold, are processed by a classifier. The work in [17] is similar, although only a threshold-based method is employed. This work considers three thresholds and, depending on the threshold that it is exceeding, the system identifies a movement (first threshold), a fall (the second threshold) or an activity of the daily living (third threshold). The work in [18] employs the accelerometer and gyroscope from the iPhone 8 to classify the collected data as fall or non-fall. The work in [19] proposes a method that combines a convolutional neural network (CNN) [20] with a long short-term memory (LSTM) network [21]. The main limitation of the approach based on a mobile phone is that, despite being very convenient when the users are outdoors, it is not very natural for individuals to carry their phone in their pockets while being indoors. There is also a line of research that detect falls based on heart rate and body temperature [22].

The second category encompasses different solutions in which the sensor is not carried by the person being monitored but, on the contrary, it is part of the environment. The work in [25] proposes the use of a non-wearable ultra-wide-band (UWB) sensor, installed in the ceiling to monitor activities underneath its area of action. Although this line of research is very promising, there are important limitations that yet prevent this solution from working in real time and under real scenarios. Thermal sensors are also proposed in [26] using infrared sensor arrays to collect data that are then processed by a recurrent neural network (RNN) model to detect when a fall has occurred. It is also worth noticing the line of research that employs the WiFi signal for fall detection. The work in [27] proposes a system for fall detection that implements a threshold-based method applied to the change in acceleration and speed of the moving object, calculated using the statistical theory of electromagnetic waves. One of the main strengths of this approach is that no special hardware is required apart from the WiFi transceiver and receiver. Although this work does not specifically evaluate user acceptance, the fact that it does not require any specific sensor or hardware is pointed out as a factor to increase acceptance. Similarly, the work in [28] also employs the 2.4 GHz WiFi band. The transmitter Tx emits electromagnetic waves that propagate in the indoor environment. These waves are reflected by the objects located in the room before arriving at the receiver Rx. A Doppler shift of the transmitted signal is caused by the person’s movement, which affects the received RF signal. Statistical features are extracted from the person’s activity, and then a machine learning classifier is employed to detect the occurrence of a fall. The works in [23,24] propose the use of a microwave Doppler sensor placed in the ceiling, emitting the microwave signal toward the floor. Falls can be detected on the basis of the pattern of the analog signal that contains the Doppler frequency.

Finally, the set of solutions based on video analysis claims, as its major asset, the avoidance of external sensors. Nonetheless, there are other drawbacks, such as the existence of blind spots, distance from the camera, especially when using Kinect, or the impact that illumination has on the performance of such video-based systems. In this sense, the use of Kinect, as a depth camera, is very common. The work in [29] presents the use of a walk-assistance robot with a built-in Kinect camera capable of detecting falls in the context of performing rehabilitation exercises. The knee angles are used for a threshold-based method to detect abnormal gait. When this happens, the system activates the robot brakes so that a fall can be prevented from happening. The use of distances to the floor, calculated using depth information, is also common. The work in [38] detects falls based on the distance between the center of the body ellipsis and the floor. A threshold-based method is implemented to detect the fall. This solution does not need to store images, so this is an important aspect when it comes to ensuring user privacy. Moreover, for being based on video, there is no need to wear or carry any sensor. The works in [33,39,40] propose the use of the distance from the head joint to the floor. The use of Kinect cameras to detect the imbalance state as a determinant of a fall is very common [30,31]. The work in [41] combines the imbalance detection approach with a sharp change in the heights of joints, as the result of a fall. In the same line, the work in [32] monitors sharp changes in any of the body-joint points. The work in [42] implements an approach based on the calculation of the body-joint speed between frames. A threshold-based approach is then employed to detect the occurrence of a fall. Similarly, the work in [43] calculates the body shape and then, the speed of movements is obtained from the variation over the frames. A machine learning classifier is employed to determine when a fall has taken place. The works in [44,45] propose the use of machine learning techniques to detect a fall from a feature vector comprised from the body-joint information. There are other approaches solely based on RGB-video analysis, in contrast to the aforementioned ones, which use the depth information provided by Kinect. The work in [34] employs computer-vision analysis to calculate the position of the user with respect to the ground to detect a fall. The works in [35,36] combine the use of environmental sensors and a RGB camera embedded in a mobile robot. The human shape is proposed in [33], as the shape orientation can also be used to determine the occurrence of a fall. The work in [37] detects a fall based on the differences of the body joints over the video frames.

Although most of the works reviewed from the state of the art focus on detecting falls independently of the factors (both intrinsic and extrinsic) that have caused the fall, there are also works that focus on factors identified as leading causes of falls. In this sense, the work in [29] points out to the rehabilitation exercises performed by people already suffering from walking disabilities. The drag-to-drop gait is identified as a major factor for falls. Similarly, the work in [46] focuses on rehabilitation exercises and, more specifically, those exercises performed with the lower limb. In this sense, the work in [47] points to any task involving gait as a major cause for falls. The works in [30,31,48,49] focus on the intrinsic factors that cause imbalances, such as muscle strength or the ability to posture control. Age is also a very common factor pointed out by many works of the state of the art, such as [14,16,23,24,33,36,41,45,50,51,52] or frailty [10], which is also related to age. Regarding the extrinsic factors, the presence of obstacles [44,53], bedtime [39,54], stair architecture design, and stair obstacles, such as the absence of a handrail, irregular riser height and an object left on stairs [49], are more commonly mentioned.

The work in [55] identifies the main post-implementation acceptance factors. The analysis of such factors can be very relevant, as they might determine the acceptance of a technological solution for fall detection. These factors are categorized into six themes: (1) concerns such as system malfunctions, false alarms, high cost, stigmatization, and lack of training; (2) experienced positive characteristics of technology, such as privacy, increased safety or unobtrusiveness; (3) experienced benefits of technology such as increased communication, increased capabilities to perform ADL, reduced burden on family or perceived need to use; (4) willingness to use technology, such as increased willingness to use or time of using (testing) technology; (5) social influence, such as influence of family or influence of organizations; and (6) characteristics of seniors, such as previous technological experience or physical environment. The analysis of the state of the art for fall detection systems shows that some works pay attention to such factors. In this sense, unobtrusiveness or being comfortable are pursued as a relevant factor for acceptance in [9,12,23,24,25,27,28,30,37,38,44,46,50,56]. Privacy and increased safety is pointed out as a user acceptance factor and pursued in [8,25,26,30,36,38,39,40,43,57]. The works in [13,48,58] concentrate on achieving a high rate of accuracy to avoid system malfunction as a cause for users’ lack of acceptance. For the authors in [29], user acceptance is determined by not having to use any external sensors, as all the required equipment is provided by the robot. Experiencing positive characteristics, such as satisfaction, is pointed out in [14]. In addition, the fact that most older adults are familiar with cell phones is stated as a factor for user acceptance as having previous technological experience [16,17,18,19]. In the overall, privacy and unobtrusiveness are the two major user acceptance factor studied by the state-of-the-art literature, although familiarity with technology is also relevant.

The analysis of the literature revision also provides relevant insights about the main limitations found by the proposed solutions for fall detection. Most of the systems have been tested with users that are not older adults or with simulated datasets, which might have biased the real accuracy of the system [9,31,46,51,57]. The expensive price of the solution is also mentioned in [25] or, related to the price, the demand of high computational resources, as stated in [40]. The technological readiness level of the solution is also a common limitation of the revised works, as some of them are not yet ready to perform fall detection in real time [12] or work in the wild [13,17,37]. The number of false positive can be an important issue, especially when actions are taken to prevent consequences after a fall has been detected [26,27,44,54]. There are a common set of limitations for approaches based on video analysis, such as blind spots, sunlight, occlusions, illumination, etc., [34,36,38,41,42,52]. The difficulty in detecting backward falls is also mentioned in [30]. The incapability of performing well when there is more than one individual in the scene or there are other moving objects is pointed out in [27,39]. The low accuracy achieved in falls occurring from beds or sofas is also mentioned in [35,38,39]. For those approaches based on mobile phones, the fact that the user has to carry the phone while being at home is pointed out as a limitation in [16]. To summarize, the major limitations found in the literature are related to the mechanisms employed to train and test, the number of false positive, the limitations intrinsic to the use of video sources, or the low accuracy when the fall occurs from a sofa or a bed.

The limitations identified in the previous work explain the lack of generalized solutions for fall detection and prevention. This research works on the hypothesis that by tackling these limitations (i.e., lack of privacy, no real-time system, the need to wear uncomfortable devices, or the cost), a solution will be achieved that potentially meets the requirements to be widely accepted by potential users: both older adults and individuals. To validate the working hypothesis, a system for fall prevention and fall detection is constructed that meets the requirements of being privacy aware, running in real time, being able to be worn in an unnoticeable manner and being low cost. Furthermore, the achieve performance should be similar or improve the state-of-the-art system performance. In this sense, this work presents a solution that comprehensively addresses falls by tackling some of the major limitations found in previous work. On the one hand, this work addresses the major factors for user acceptance as they are unobtrusiveness and privacy. It has to be noticed that, whereas video-based approaches cater for unobtrusiveness, they fail to do so for privacy, even when only depth information is used, as with Kinect cameras. Nonetheless, the use of wearable devices has also been considered as an unobtrusive solution, especially when the wearable sensor is light, small and elegant. This work therefore proposes a solution based on a wearable sensor, using the Puck.js sensor, embedded in a comfortable waist hanger design. On the other hand, this proposed approach addresses one of the major causes of falls, the bed-exit action. This operation is, at the same time, one of the major limitations of state-of-the-art works. In order to do so, the proposed approach combines information from the wearable sensor with information obtained from an environmental sensor located underneath the mattress that determines the presence and location of the individual on the bed, based on pressure information. The main advantages of this solution, compared to the state-of-the-art solutions, is that it works in real time and, more importantly, it caters to the unobtrusiveness and privacy levels required for ensuring user acceptance.

This article is structured as follows. First, the materials and methods used for this research are described, paying special attention to the proposed architecture that enables the system to run in real time and at low cost. Then, the experimental validation and the obtained results are described in Section 5. The obtained results are discussed in the next Section 6, and the main conclusions drawn from this experiment and future work are summarized in Section 7.

## 3. Proposed System

Falls in older adults are a public health concern worldwide. As the population ages, this problem is set to increase. In order to prevent or alleviate the problems associated with falls, technological systems based on sensors are emerging as a way to process information in real time to determine the occurrence of a fall. These systems can be divided into two major categories [59]:Fall detection systems: These systems are intended to use one or more sensors to detect, in real time, whether a person has suffered a fall. In general, inertial sensors are usually employed, consisting of accelerometers and gyroscopes that can measure abrupt movements. However, there are also other types of sensors that are capable of detecting impacts. These systems are the most common ones and the ones that have received most attention from the scientific community. However, it is important to note that these systems cannot prevent the fall, but rather their main responsibility is to mitigate the negative effects of the fall.Fall prevention systems: These types of systems are used less frequently and they do not give immediate results like the previous ones. The sensor technology involved is similar to that used in fall detection systems, except that these systems focus more on analyzing and extracting features about the user’s gait, posture or behavior.

This work addresses the fall problem in older adults from these two perspectives. From the perspective of fall detection, the aim is to automatically detect falls in real time, as this reduces the negative consequences, both physical (abrasions, dislocations, and broken hips) and psychological (fear of falling syndrome). From the perspective of fall prevention, this work aims to enable mechanisms prior the fall occurrence. In the context of a nursing home or a person with a caregiver at home, the individual will generally be supervised and accompanied at all times during the day. The most problematic time is at night, when there is less supervision to guarantee a good night rest. This time of day is particularly sensitive for people suffering from conditions such as dementia or reduced mobility, as bed exit increases the risk of falls. Traditionally, restraints have been used to prevent these unattended bed-exit attempts to prevent falls in individuals that suffer a severe condition. These restraints, beside being uncomfortable for individuals, represent a loss of freedom, which is why significant efforts are being aimed at finding alternative mechanisms to avoid them.

This work also contributes to the elimination of restraints by providing an unobtrusive mechanism to monitor individuals so that whenever an intention to exit is detected, the caregiver can be automatically notified and act in a timely manner.

Sensors, both environmental and wearable, are proposed to address both of the above perspectives. These sensors collect information about the individual that will help caregivers and therapists to provide the best possible care, preventing falls from occurring. This solution mainly consists of a processing node, which in our case is a Raspberry Pi 4, and two different types of sensors that are used to monitor the older adult: an inertial measurement unit (IMU) and three pressure sensors. The following subsections provide the details of fall detection and prevention systems proposed here.

### 3.1. Fall Detection

The purpose of the fall detection system is to detect a fall as early as possible such that help and assistance can be requested in order to minimize the negative consequences of the fall. The system proposed here improves an earlier version presented in [60], based on the MetamotionR inertial sensor manufactured by Mbientlab. This sensor gathers measures from an accelerometer, a gyroscope and a magnetometer unit. Both the previous and current systems follow the same approach of extracting various features to conclude, based on a previously trained model, whether the individual has suffered a fall or not (video illustrating the use of the fall detector: https://youtu.be/8Zn43OqBd1w (accessed on 5 April 2022)).

The following subsections describe both the hardware and employed method proposed for fall detection.

#### 3.1.1. Hardware

The proposed solution for fall detection implements a wearable-sensor-based approach. More specifically, it employs a Puck.js sensor, which is an inertial sensor with a set of components as listed underneath:Magnetometer: This sensor is used to measure magnetic forces. It can also measure variations in magnetic fields.Accelerometer and gyroscope: These two sensors provide inertial measurements, such as acceleration and rotation, measured in g and degrees per second, respectively. These measurements are essential for fall detection, as they are a strong indicator of abrupt movements. This sensor has some advanced features built upon the raw data obtained from these two sensors, such as step counting or sensor tilt.Light sensor: This simple light sensor gives a value between 0 and 1, indicating the presence or absence of light.Temperature: It provides the ambient temperature measured in Celsius degrees.

In addition to these components, advanced functionalities are constructed upon the information provided by these sensors, such as step counting or tilt detection. Nonetheless, the most attractive feature is the possibility it offers to modify the behavior of the sensor itself. Despite the wide variety of IMUs that can be found on the market, they generally offer a closed solution, with data only being accessible through close APIs. On the contrary, the Puck.js sensor offers a firmware that provides an extra layer with a Javascript interpreter for microcontrollers called *espruino*. This enables the customization of the sensor behavior, as well as, for example, disabling components that are not needed in order to save battery life. However, despite all these positive points, the inclusion of this extra layer entails an overload due to the use of the interpreter, which has an impact on the sensor sampling frequency, reducing it to a peak frequency of 50 Hz. This would be, however, sufficient according to the work of [61], given that the human body movements are considered to occur in the range of 0 to 15 Hz. In fact, according to the Nyquist’s theorem [62], the sampling rate could be reduced to around 30 Hz. Furthermore, the data collected from the sensor are then sent to a processing node, which in this case is hosted on a Raspberry Pi 4. This node consists of a SoC with four Cortex A-72 processing cores, with up to 8 GB of RAM, 2.4 and 5 GHz IEEE 802.11ac wireless connectivity, as well as both Bluetooth and BLE connectivity.

#### 3.1.2. Architecture

The Puck.js sensor is therefore proposed to obtain the data related to the person movement and, consequently, to be able to determine when a fall has occurred. In order to do so, it is necessary to provide a service that supports Bluetooth communication at the processing node, since the sensor communicates its information via Bluetooth low energy (BLE). The main challenge to be addressed is the short range of Bluetooth communication in this kind of device. To maximize battery life, these devices are quite power constrained, and therefore, Bluetooth class 2 is used, which limits the power to 2.5 mW and the range to around 5–10 m. It is difficult to maintain a continuous communication with the sensor due to these limitations, and there may be data loss when streaming data to the processing node. A possible solution would be to use a class 1 device, increasing the range to 100 m, although this is unfeasible, as it would increase the power to 100 mW, raising the power consumption and therefore reducing the battery life of the sensor.

This work avoids continuous communication, both because of the possible loss of information and because of the increase in energy consumption that this implies. Thus, the use of BLE announcements is exploited.

The possibility offered to modify the firmware of the Puck.js sensor has made it possible to deploy a threshold-based algorithm (TBA) along with a state machine devised to determine the movements that are likely to be a fall. When this happens, the sensor stores in its internal memory the data of the 1 s time window corresponding to such event. A certain movement is considered a fall when the signal vector magnitude (*SVM*) ($SVM=\sqrt{\left(|A_{x}{|}^{2}+|A_{y}{|}^{2}+|A_{z}{|}^{2}\right)}$ ) of the acceleration is greater than 2.5 G. For further details of the state machine, please refer to  [60]. The sensor publishes an announce alerting about the possibility of a fall event. This announce is received by a processing node (Raspberry Pi). Whenever such a notification is received, the processing node connects to the sensor and downloads the data stored in the sensor internal memory.

Upon receiving the event and downloading the data, the feature extraction service that runs outside the sensor performs a data pre-processing so that the movement data are represented in a more appropriate way, and their dimensionality is reduced. The features yielded by this service are the following:Acceleration SVM mean;Acceleration SVM variance;Acceleration Y mean;Acceleration Z meanAngular velocity Y mean;Angular velocity Z meanFall time;Acceleration Y standard deviation;Acceleration Z standard deviation;Angular velocity Y standard deviation;Angular velocity Z standard deviation;Fourier coefficient in acceleration SVM;Fourier coefficient in angular velocity SVM.

These features will be sent, along with the MAC of the sensor that has captured the data, to a multicast group to which the fall detection service is listening. This service consists of a previously trained model that, based on the features, will classify the movements into a fall or an ADL. In the event of detecting a fall, this service will alert a designated person so that he/she can assist the victim immediately. Box 1 shows the pseudocode for the state machine implemented in the firmware of the IMU itself, where, after detecting an acceleration higher than the threshold, the obtained measurements are stored in a window of one second. If, once this window is saved, a peak is detected again, this may mean that the user may be doing some kind of physical activity (for example, running), so the stored window is removed. Otherwise, if after storing the window the acceleration remains constant and no acceleration higher than the threshold is obtained, it may mean that the user has suffered a fall, so a fall event is sent. Box 2 shows the pseudocode of the intermediate and final services that receive the data collected from the sensor, where once a fall event is received, the data are downloaded, its features are extracted, the data are normalized, and the previously trained model is used to classify the activity as a fall or ADL. If a fall is detected, an alarm is triggered, alerting a caregiver.

Listing 1IMU firmware pseudocode for fall detectionstate = SAMPLING; while(1) {     acc = readAccelerometer();    angular_velocity = readGyro();     acc_svm = sqrt(pow(acc.x, 2), pow(acc.y, 2), pow(acc.z, 2));     switch(state) {        case SAMPLING:            if(acc_svm > 2.5) {                state = POST_PEAK;                time_reference = time.now() + 1000;            }        case POST_PEAK:            log.save(acc);            log.save(angular_velocity)             if(time.now() > time_reference) {                state = POST_FALL                time_reference = time.now() + 1500;            }         case POST_FALL:            if(acc_svm > 2.5) {                log.clear();                state = POST_PEAK;            }             if(time.now() > time_reference) {                state = SAMPLING;                send_event(FALL_EVENT);            }    } } 

One of the main advantages of the proposed architecture is that, as the services that obtain the sensor data via BLE communicate with the fall detection service using multicast groups, these services can either be on the same processing node or on a separate one connected to the same network. Figure 1 depicts the different elements that comprise the proposed solution for fall detection as well as the workflow from the user fall until the alert is triggered. The *IMU2MCAST* service is the one listening for the sensor events. Upon receiving an event, the data are downloaded by the *Download Service*, which is provided with the MAC of the sensor that sent the event. These data are then sent to the *Feature Extractor*. The extractor obtains from the raw data, a set of features that represent the user movement, which are sent back to the *IMU2MCAST* service. Finally, *IMU2MCAST* forwards the features to a multicast group to which the *Fall Detector* is subscribed, processing the features and taking actions depending on whether the set of features is classified as a fall or an ADL.

Listing 2Gateway pseudocode for fall detection.// Intermediary servicesevent, mac = imu2mcast.wait_for_event();if(event == FALL_EVENT) {    data = download_service.download_data(mac);    features = feature_extractor.extract_features(data);    imu2mcast.send(features, FALL__DETECTION_MCAST_GROUP);} //Fall detectorfeatures = wait_for_features();features_norm = normalize_features(features);fall = model.predict(features); if(fall) {    raise_alarm();}

### 3.2. Fall Prevention

Far more important than detecting falls is preventing them from occurring in the first place. To this end, a wide variety of systems have been devised to address such a complex topic. The system presented here focuses on monitoring people at night, as this is when they tend to be less attended and more vulnerable. The proposed system provides information when an older adult, being monitored, has to exit the bed or is about to do so, thus alerting the caregiver about this situation. This system is very useful for people suffering from dementia or any other cognitive impairment as well as with reduced mobility. For people suffering such conditions, just transiting with no supervision entails a high probability of suffering a fall. It is, therefore, especially important to be able to detect when these people exit the bed by their own means and without help (video illustrating the use of the fall prevention system: https://youtu.be/–QLtgktoDw (accessed on 5 April 2022).

The following subsections provides the details, both in terms of hardware and methods, that comprise the proposed solution for fall prevention.

#### 3.2.1. Hardware

There are two different sources of information that are employed in order to detect if the person is on the move. In the first place, we take advantage of the IMU already in use for fall detection, the Puck.js. Recall that the behavior of this sensor is completely modifiable, i.e., we can modify the sensor firmware to send, in addition to the fall event previously discussed, a movement event that indicates that the user has stood up or is walking.

On the other hand, the system focuses on bedtime monitoring, so pressure sensors are used to determine whether the user is in bed or not. More specifically, three 600 mm film pressure sensors are used, which change their electrical resistance when force or bending is applied to the sensor membrane. These sensor strips will be placed under the mattress, indicating whether someone is lying on the mattress based on the value provided by the sensors. Even the movements of the user on the bed are taken into account. The processing node for this system is the ESP32-DevKitC board, which is equipped with both WiFi and Bluetooth, with 4 MB flash memory and an ESP32 SoC. This board was mainly chosen because of its good results for IoT solutions, its low cost, having both Bluetooth and WiFi connections and the possibility to use several analog inputs.

The prevention system therefore consists of the IMU Puck.js previously explained and an ESP32 node which will have the pressure sensors connected to three analog inputs, also using a voltage divider with a 1 kΩ resistor. The connection between the ESP32 node and the sensors is to obtain a higher value from the inputs as more pressure is exerted on the sensors (which are nothing more than a variable resistor). Figure 2 shows the schematic diagram for each analog input. The three sensors had to be placed across the bed width, as a sleeping person will mostly move horizontally across the mattress, and it is these movements that can give significant clues as to whether the user is going to exit the bed. Given the bed size used as a test bed for this system, which is 90 cm wide and 190 cm high, the three sensors were placed as follows: one sensor in the center at 45 cm from both side edges, and the other two sensors at the ends of the bed, each at a different edge and more precisely at 20 cm from the edge of the bed. Regarding the bed height, using the headboard as a reference, the beginning of the strips was placed at 57 cm from the headboard so that the strips were placed just under the torso of the subject, as this is the area that exerts the most pressure on the bed when resting on it. The sensors with their respective location on the underside of the mattress can be seen in Figure 3. The ESP32 processing node with its required set-up can be seen in Figure 4.

#### 3.2.2. Architecture

The proposed system architecture follows a twofold approach. On the one hand, motion detection is used to determine the user’s intention to exit bed and, on the other hand, the information provided by the pressure sensors is used to detect when the person has actually exited the bed.

The motion detection logic is implemented using the accelerometer unit provided by the IMU sensor. For this purpose, the value of the X and Z components of the accelerometer are continuously monitored. These components provide information about the acceleration of the body. Based on the tests carried out, it was concluded that the standing up action can be considered when the X component of the accelerometer exceeds the value of 0.9, and the Z component is between the range of 0.4 and −0.4. The use of these two components prevents false positives from occurring when the user lies down. Whenever these values are reached, the sensor triggers a motion event using the BLE announcements in the same way that fall events are triggered. Considering there is already a software service for fall detection in place that receives events via BLE and forwards them to multicast groups (*IMU2MCAST*), this is reused for this purpose as well. However, in this particular case, the service directly forwards the event to the corresponding multicast group, obviating the step of downloading the data from the sensor as in the case of fall detection. In Box 3 and Box 4 the pseudocode for the IMU’s firmware and the gateway’s software for fall prevention can be observed.

Listing 3IMU firmware pseudocode for fall prevention.while(1) {    acc = readAccelerometer();     if(acc.x > 0.9 and (acc.z > -0.4 and acc.z < 0.4) {        send_event(MOVEMENT_EVENT);    }}

Listing 4Gateway pseudocode for fall prevention.event = imu2mcast.wait_for_event();if(event == MOVEMENT_EVENT) {    imu2mcast.send(event, FALL_PREVENTION_MCAST_GROUP);}

On the other hand, the logic that determines whether an individual is in bed is based on the use of three pressure sensors located under the bed mattress and connected to the ESP32 board. These sensors have the advantage over the IMU that they can be connected directly to a WiFi network, and can communicate data using a client/server paradigm via sockets. Although it was initially considered to add a calibration period so that the sensors would not consider the pressure of the mattress, first tests yielded that pressure exerted by the mattress is negligible and it is not even detected. The proposed module therefore processes the information from these three sensors located underneath the mattress, generating information in the form of events, as described below:NO_PRESENCE: Whenever there is no pressure exerted in any of the sensors. In general, this situation occurs whenever all pressure sensor reads are 0 or close to 0.SITTING_IN_LEFT_EDGE and SITTING_IN_RIGHT_EDGE: This represents a person sitting on one of the bed sides, which may be a strong indication that the user is about to exit the bed. This event is considered whenever the sensor on one of the sides yields a value of 0, and the value of the opposite sensor is significantly greater than that of the sensor in the middle.LYING_IN_RIGHT_EDGE and LYING_IN_LEFT_EDGE: This occurs when the person is lying close to one of the two bed sides. In this case, the sensor placed in the middle yields the highest value among the three sensors. The difference between the middle sensor with respect to one of sides is less than the difference between the two sensors placed in both bed sides.LYING_IN_MIDDLE: The person rests in the middle of the bed. In this case, the middle sensor has the highest value among the three sensors, and the differences between the middle sensor and the bed-side sensors are greater than the difference between the values yielded by the two bed-side sensors.

Box 5 shows the pseudocode for the algorithm devised to detect the user position on the bed.

Listing 5Bed presence algorithm pseudocode. middle = readSensor(MID);left = readSensor(LEFT);right = readSensor(RIGHT); if (middle near 0 and left near 0 and right near 0) {    presence = NO_PRESENCE} else if (middle > right and middle > left and diff(right, middle) > diff(left, right) and diff(left, middle) > diff(left, right)) {    presence = LYING_IN_MIDDLE} else if (left > right and left > middle and right near 0 and diff(left, middle) is large) {    presence = SITTING_IN_LEFT_EDGE} else if (left > right and middle > left and diff(left, middle) < diff(right, left) {    presence = LYING_IN_LEFT_EDGE} else if (right > left and right > middle and left near 0 and diff(right, middle) is large) {    presence = SITTING_IN_RIGHT_EDGE} else if (right > left and middle > right and diff(right, middle) < diff(right, left) {    presence = LYING_IN_RIGHT_EDGE} else {    presence not recognized} send_event(presence);

It should be noted that when the user changes position in bed, the detection process becomes more complicated. This is because the data produced in the transitions do not follow clear patterns. To alleviate this problem, two actions are implemented. On the one hand, if a detected position does not follow any known pattern, it is discarded, and no change of position is considered to have occurred. On the other hand, to avoid a position change from being identified as an actual posture, it is stated that the recognized pattern should last longer than one second. In either case, the events are sent via a TCP socket to a server that can either be hosted on the same processing node (Raspberry Pi) that communicates with the Puck.js sensor or on a completely different node, as long as the sensor and the node are in the same network. The server, named *BedPresenceService*, receives the event from the sensor, and in turn forwards it to a multicast group in which those services that want to obtain information about the bed presence status are listening. Box 6 shows the pseudocode of this service.

Listing 6BedPresenceService pseudocode.presence_event = wait_for_event();send_event(presence_event, BED_PRESENCE_MCAST);

Finally, there is a third service, named *ActivityDetector*, whose responsibility is to subscribe to the multicast groups in which announcements are published about motion and bed position events. Thus, the *ActivityDetector* combines both sources of information to conclude whether there is a real risk of a fall, in which case the caregiver is alerted. There are therefore two sources of data that provide independent information; one tells whether the user is on the move and the other one tells the user position on the bed. The two sources of information can be combined to conclude whether the user is trying to get up. The scenarios considered risk situations that this service will inform about are the following ones: (1) no presence has been detected on the bed, and (2) the user has been detected sitting on the bed side and there is movement. In the second scenario, two different information sources are combined to reduce false positives. Box 7 shows the pseudocode for the *ActivityDetector*.

Listing 7ActivityDetector pseudocode.presence = recv_presence_event();movement = recv_movement_event(); if(presence == NO_PRESENCE) {    raise_alarm();} else if((presence == SITTING_IN_RIGHT_EDGE || presence == SITTING_IN_LEFT_EDGE) && movement == True) {    raise_alarm();}

Figure 5 summarizes the fall-prevention system workflow. When the IMU detects a movement with an acceleration in the X component greater than 0.9 and an acceleration in the Z one between 0.4 and −0.4, it triggers a movement event to the *IMU2MCAST* service, which forwards the data to the corresponding multicast group. Meanwhile, the sensor that continuously monitors the bed sends the current state of bed presence to a server, named *BedPresenceService*, which forwards the information received to another multicast group. Finally, the *ActivityDetector* service is in charge of processing the information from both sensors. This service subscribes to the multicast groups for movement and bed position detection in order to receive the required information and to process it. When the system detects one of the two scenarios explained before, the service considers that the person is trying to exit bed.

### 3.3. A Comprehensive Solution for Fall Detection and Fall Prevention

Both the fall detector and the fall prevention system have been so far described separately. Nonetheless, the main objective of this work is to present a comprehensive system that can monitor an individual for an entire night so as to alert a caregiver when circumstances arise that pose a risk to the individual, such as falling or exiting the bed. Figure 6 provides an overview of how the system works and how the various components of the system interact.

The processing can be clearly divided into three stages. The first stage involves the sensor system in charge of processing the data on the edge, following an edge computing approach. In this case, the sensors used are the IMU Puck.js to obtain the inertial measurements of the individual and the ESP32 connected to the three pressure sensors that detect the user position on bed. Then, there are a set of intermediate services in charge of communicating with the sensors and receiving the information they process. Their function is to forward the data to the upper services that will make the decision, as well as, in some cases, performing some processing on the data. The information received and processed is sent to multicast groups to which the services that require them are subscribed to. Finally, we have the set of services that, thanks to the intermediate services, perform additional processing to determine whether it is necessary to alert the caregiver about a risk situation.

## 4. Method

This work seeks answering the research question of how to prevent and detect falls in older adults, using a solution that is widely accepted and adopted by older adults and their caregivers. The proposed solutions need to, therefore, address the aspects that impact user acceptance, as justified in the previous work section, and are as follows: being privacy aware, working in real time, avoiding the need to carry uncomfortable devices, and having a constrained price. In this sense, this work proposes to build a solution that caters to these aforementioned requirements plus minimizes the false negative rate and achieves similar performance as the best solutions found in previous works. Based on the proposed system described in the previous section, two methods are proposed for fall detection and fall prevention that, when combined, comprehensively address the goal of preventing and detecting falls in older adults. This section describes how these two methods were constructed, whereas the following section describes how these were experimentally validated.

### 4.1. Fall Detection

There are several aspects that were considered in the methodology proposed. The literature is very diverse with regard to where to locate the sensor in order to detect falls. According to [63], the most popular location, which has been proven to be most effective one for fall detection, is the waist, although the use of the head and chest may also be feasible. Based on this premise, it was decided to place the sensor at the waist, as shown in Figure 7. Moreover, the sensor is placed with the orientation shown in Figure 8 such that the X-axis is perpendicular to the ground and upwards, the Y-axis is perpendicular to the user’s body and to the left, and the Z-axis is parallel to the ground pointing to the user’s body.

The main challenge of fall detection systems is to differentiate whether a movement corresponds to a fall or an activity of daily living (ADL). In order to do so, the proposed method relies on a trained model, and from a set of user movement features is able to recognize whether the movement corresponds to a fall or an ADL. ADLs are those everyday movements of the user that, in some cases, produce fall-like data, such as running or sitting.

This model was trained on data captured from a set of users performing both ADLs and different types of falls. More specifically, five ADLs (hit the sensor, jump, run and stop, sit on a chair, and pull the sensor) and four different types of falls (forward fall, backward fall, fall on the right side, and fall on the left side) were performed.

These data were collected in a lab context (in the Institute of Information Technologies and Systems, University of Castilla-La Mancha). Seventeen different people performed the exercises, comprising four women and thirteen men, with an average age of 30±8.02 years, an average height of 174.18±7.85 cm and an average weight of 74.35±9.71 kg. To perform falls, a mat was employed to soften falls but it still let them simulate reality as closely as possible.

The support vector machine (SVM) classifier, one of the most widely used in fall detection [64,65,66], was chosen, yielding better results than the KNN classifiers [67].

The training phase of the model was followed by the testing phase, for which two different sets of tests were conducted. During the first set of tests, the dataset collected from the 17 people were split into a training and a testing set. The testing set is used to predict the type of event (either fall or ADL) and calculate, afterwards and based on the obtained results, the accuracy of the system and make improvements to the model. In this sense, a subject was monitored during a full night while sleeping. Throughout the night, data were captured on those movements that are likely to be falls, and the prediction yielded by the model was stored. In cases of incorrect prediction, the data were used to retrofit the model. This was done only once to avoid overfitting. Once this information was added to the model, the second test set was carried out, which consisted of a series of exercises performed by a new set of users. These exercises comprised the following:Jump;Run and stop;Sitting on a chair;Forward fall;Left side fall;Right side fall;Backward fall.

### 4.2. Fall Prevention

A series of tests were carried out in order to observe the behavior of the different sensors when the user performs different actions and to obtain a measure of the system performance. These tests were split into two groups: those involving the IMU and those involving the bed presence sensors. The tests were carried out by a user performing common activities.

A threshold-based algorithm (TBA) was proposed for movement detection based on the inertial information provided by each of the axes of the IMU. This algorithm will analyze the sensor inertial measurements when the user attempts to exit the bed. To evaluate the proposed TBA, a person simulated a set of activities, as known: exit the bed from the left and right side of the bed, and then, in both cases, also walk away. The data obtained from the inertial sensors and, more specifically the data from the different components of the accelerometer, were analyzed while performing these activities. Figure 9 and Figure 10 show the accelerations of the components when performing a bed exit from the left and from the right sides, respectively. For both actions, a common factor can be observed, in which X starts close to 0 and Z close to 1. Then, as the user gets up, it can be observed how this situation is reversed. When the user is standing up completely, the value of the X component oscillates around 1 and 0.9, and the value of the Z component is between 0.4 and −0.4. These values are to be expected since the X component corresponds to the axis perpendicular to the ground and the Z component is the parallel to the ground. The value of the Y component is ignored, as it does not provide much more information beyond what is already provided by the X and Z components. The Y component would be relevant to identify whether the user exits from the right or left side, which, for our purposes, is irrelevant.

Furthermore, the study conducted to determine the presence in bed using the pressure sensors was mainly based on the data collected from static positions in the bed. An analysis of the pressure readings was carried out on the basis of the user position, in order to learn a behavior pattern for these readings, according to the posture adopted in the bed. A summary of the activities and the results obtained for each of the sensors can be seen in Figure 11.

Expected results were obtained depending on the position, which indicates that the placement of the sensors is optimal for recognizing the user actions. Those positions in which the user is lying in the middle yield measurements in which the central sensor obtains much higher values than the value of the other two sensors, both of which report similar values. Those positions in which the user is lying on one side of the bed produce values that are high and almost equal for both the central sensor and the sensor positioned at the bed side where the patient is lying, with the measures of the central sensor being higher. The sensor at the opposite side of the bed produces negligible measures. Finally, when the user is sitting on the bed, very high values can be observed only in the sensor located at the side of the bed where the user is sitting, while the value of the other two sensors are very low. This facilitates the recognition of this activity, and the fact that the user is sitting on the bed side is a strong indication that he/she is about to exit the bed.

## 5. Results

The system proposed here was experimentally validated. For this purpose, a set of experiments was designed to validate the fall detection system, on the one hand, and the fall prevention system, on the other. The following sections will describe and analyze both the proposed experiments and the obtained results.

### 5.1. Experiments and Results Validating the Fall Detection System

The validation of the fall detection system faces a major limitation in that it is not feasible to test the proposed approach with the target user group (i.e., older adults). Simulating falls is a high risk activity and for this reason, it is discarded to record a testing dataset involving older adults. In addition to this limitation, falling is not a natural activity for people, like walking or exiting the bed. Thus, testing the system with a simulated-fall dataset does not ensure that the system performs equally well in a real environment with real falls. Simulated falls may not include instances of real falls, and that might bias the system performance. Given these baseline conditions, it was decided to experimentally validate the system with 11 users, 6 of whom took part in the exercises carried out to train the system, whereas the other 5 had not previously performed the exercises.

The testing dataset consisted of each subject performing seven different exercises, three of which were ADLS (jumping on site, running and sitting on a chair) and four of them were different types of falls (front fall, right-side fall, left-side fall and back fall). The exercises were performed twice by each subject. The testing dataset was provided to the classifier to determine which of them were falls. The results obtained are shown in Table 2.

The obtained results show that, in the overall, the majority of the tests were correctly classified. Nonetheless, it is worth noting that the classifier fails to recognize the backwards fall up until five times. The literature revision already pointed out the difficulty of recognizing this type of falls, which is consistent with the obtained results. This results are also explained because of the difficulty of performing such a type of fall as well as its possible resemblance to the sitting activity. Regarding the rest of the exercises, the results are adequate, with the left and right falls failing to classify one exercise each. Three false positives are obtained, one for the sitting activity and two for the running activity.

The system performance is calculated from the results obtained from the testing dataset. To this end, we calculated the accuracy (Accuracy=(TP+FP)/(TP+FP+TN+FN), the specificity (Specificity=TN/(TN+FP)) and the sensitivity (Sensitivity=TP/(TP+FN)), obtaining the following results:Accuracy: 93.51%Sensitivity: 92.04%Specificity: 95.45%

In general, accuracy is usually the most intuitive metric for testing the performance of a system, but in many cases, it can be misleading, for example, if there were few positive cases in the tests, and the system classified all cases as negative. The accuracy would be acceptable, despite not having classified the positives correctly. In order to avoid this problem, the use of other metrics, such as sensitivity and specificity, can be quite useful, as both metrics represent, respectively, the number of positives and the number of negatives correctly classified. The high specificity obtained by the system is worth mentioning because it means that most activities that are not falls are correctly detected. The main drawback is the high rate of false negatives, caused by backwards falls. In geriatric care, high sensitivity is more important than specificity since it is preferable to have false positives for falls than to have unreported falls.

It is important to note that the system training, in addition to the predefined exercises, also include a period in which a user was monitored and data were collected at night. This ultimately resulted in the system having more information on those movements that are not falls, and therefore contributed to a better specificity. Future work is still needed to improve the sensitivity, key for fall detection, by collecting more fall data.

### 5.2. Experiments and Results Validating the Fall Prevention System

The fall prevention system was tested in two parts, as it is comprised of two subsystems: the motion system and the bed presence system. Future work remains to be done on testing the prevention system as a whole.

#### 5.2.1. Validating the Bed Presence Detection System

In order to test the bed presence system as well as the algorithm that determines the position in bed, whenever presence is detected, two different tests were conducted. The first tests consist of performing various movements in bed during which the individual changes from one position to another, for example, from lying in the middle to sitting on the right side of the bed. Sequences are also added in which the individual performs the complete task of exit bed and lying down on bed. The second tests are more challenging for the system, with a sequence of actions in which users go through all possible positions in the bed. This second tests are therefore more relevant in terms of the obtained results, and they are the ones to which we will refer to when discussing the results.

The first set of tests shows, first of all, different movements performed by a user without any mobility problems. The results of this test can be seen in Table 3, which compares the expected result with the obtained one. In green, correctly classified postures are indicated. In red are those that were not correctly detected, and in yellow, those intermediate positions that were not fully detected but obtained results similar to those expected. It is worth noting that the majority of both initial and final positions are correctly detected, and there are only deviations in the results of two intermediate positions and one final position. As for the intermediate positions, it can be observed that both errors are similar, not detecting a posture of lying on the side when the individual lies down. This may be due to the fact that user moves faster than expected and does not actually reach that position. This results in the system not being able to detect this position, thus detecting lying middle instead. As for the misclassified position, it can be observed that it confuses the lying middle position with lying right, this may be due to the fact that the lying right, lying middle and lying left positions are very similar and tend to be confused by the system. Furthermore, in addition to the tests with the aforementioned user, two sequences were carried out with a real user, an older adult living in the El Salvador nursing home, located in Pedroche Córdoba, Spain. This was done so that the system could be tested with the target audience for which this type of system is intended. It is expected that the movements of this group of users will be slower. The results can be seen in Table 4, which shows that all changes of the user positions are correctly detected. It is expected that the system performance improves with older adults, who generally have mobility problems and will make slower movements where pressure changes are easier to observe.

The second set of tests consists of 11 different people, most of them young people, performing all possible positions on a bed. This set of test is intended to better detect the shortcomings of the system, as these users will move faster than the target audience. This set of tests consists of a sequence of actions in which the following positions were performed in this order: exit bed, sitting right, lying middle, lying right, lying left and sitting left. In order to evaluate the classification performed by the algorithm, a confusion matrix was created. In this, it is possible to observe how many of the 11 users have their positions well classified, and in the case of failure, to know which position they were confused with. This confusion matrix can be seen in Table 5.

The analysis of the matrix shows that, in general, almost all the positions are correctly detected for all the users. Only two of them are erroneous. The first error affects the lying right position, which in one occasion is detected as lying middle. This error can be explained, as the first tests concluded that these positions were very similar and, therefore, easy to confuse. There is not a reasonable explanation for detected errors with the sitting left posture, which was detected erroneously in 45% of the cases, which is unexpected, as the algorithm detects the sitting right position correctly in all situations. There could be two explanations for this fact: either the algorithm does not detect sitting positions correctly when the user exits the bed, or the left sensor for some reason did not perform properly or was misplaced the day of the tests. The data collected generally show that the readings of the left sensor are much lower than those of the right sensor in symmetrical positions such as sitting right and sitting left, where both sensors should give similar values, respectively. This may be an indication that in positions in which the left sensor is supposed to give higher values, smaller values are obtained, causing the system to confuse the sitting left and lying left positions since the higher values enable these two postures to be differentiated.

Considering that a multi-class classification is being performed, it is difficult to analyze accuracy, sensitivity and specificity for all classes in general when studying the metrics, so it was decided to perform this analysis on a class-by-class basis first. Table 6 shows the obtained metrics for each class. The analysis of these metrics reports that, in general, good results are obtained, with each of the metrics scoring close to 100% in almost all positions. The only posture that gives problems in the classification is sitting left, with a sensitivity of 54.24%. This indicates that the system does not detect postures where the user is actually sitting on the left side of the bed, suggesting that the detection of this posture should be improved to increase the rate of true positives. As it can be noticed in this case, the accuracy metric can be somewhat misleading, since despite obtaining very good results regarding the accuracy of the sitting left posture, from the sensitivity metric, we can see that the detection rate of true positives is very low and needs to be improved. The average of each of the metrics is calculated to estimate the performance of the system, considering the result for each of the postures.

#### 5.2.2. Validating the Motion Detection System

Regarding the motion detection system using the IMU as part of the prevention system, a set of tests was also developed to determine the accuracy of the TBA, designed based on observations. This set consists of 40 tests performed by a single user executing the bed exit action. Out of these 40 tests, 20 were performed on the right side of the bed and 20 on the left side of the bed. Given that the bed exit action can be divided into different stages, the analysis of the results is segmented into three different phases. The phases are: lying–sitting, sitting–standing and standing–walking.

The ideal situation is to detect motion in the first phase because the individual has not yet stood up. As the individual progresses through these phases, the risk of falling increases. Furthermore, this is related to the *ActivityDetector* service, which considers that the user has exited the bed when it detects that the user is sitting on the bed side and there is movement, so the main interest is in detecting movement before the user is fully up.

It is important to note that these tests cannot produce false positives, as they are very limited activities in which only TP or FN is expected; because of these limitations, sensitivity is the metric to use, telling how many positive measures are truly detected. Hence, it is necessary to conduct another experiment to verify the system ability to avoid this type of situation, in which we calculate the number of negative samples classified correctly. The test results for each of the phases would be a sensitivity of 92.5% for lying–sitting, 100% for sitting–standing and 100% for standing–walking. The lying–sitting phase obtains the lowest sensitivity rate with 92.5% of the cases being correctly detected. It is followed by the sitting–standing and standing–walking phases with 100%, both with consistent results where all user movements are correctly detected. The obtained results demonstrate that user movements can be detected in early phases of the bed exit action, which, therefore, improves the chances of providing early assistance.

To study the case of false positives (FP) in motion detection, the tests were divided into two parts. In the first part, 40 tests were performed in which the user moved along the bed, from lying on the right side of the bed to lying flat with the back resting, and then to lying on the left side of the bed. These movements were repeated 4 times for each test; none of these 40 tests resulted in a false positive. In the second part, to test the system under more realistic circumstances, a user was monitored for several nights. The user was monitored for false positives under real-life circumstances while sleeping. During this monitoring period, again, no false positives were detected. Given these results, it can be concluded that the system is robust enough to avoid false positives while the user is lying down, implying a specificity rate of 100%.

Based on specificity and sensitivity, the accuracy can be calculated with the formula Accuracy=Sensitivity∗Prevalence+Specificity∗(1−Prevalence). In this case, the prevalence would be calculated as the number of true positives divided by the number of samples in our test battery (Prevalence=P/P+N), thus yielding 40/40+40=0.5=50%. It should be noted that with respect to the negative samples, only those corresponding to the 40 tests performed under control were taken, and not those under real circumstances, since in the last ones, it was impossible to know the number of samples. The prevalence data can be used to calculate the accuracy; for example, considering the sensitivity for the lying–sitting phase, the accuracy is Accuracy=0.5∗0.925+0.5∗1.0=0.9625=96.25%. All metrics can be observed in Table 7.

Finally, it should be noted that these tests were carried out on young people, so it is possible that the results may be different when used with older adults, who perform movements more slowly.

#### 5.2.3. Validating the Comprehensive System Fall Prevention

While no specific tests were performed for the comprehensive fall prevention system, as it simply combines the information of presence in bed system and user motion system, and these systems are separately tested, it is possible to estimate the expected accuracy of the prevention system from the inner systems. Risk situations occur when user activity is detected, and (1) there is no presence in the bed, or (2) the user is sitting on the bed edge and movement is detected by the IMU.
No bed presence: In this first case, there is only one source of information, and that is the information obtained from the bed presence service. The accuracy of this assumption will therefore be the accuracy of the bed presence service in detecting that there is no user in the bed. This information is obtained by referring to the metrics of Table 6, which is the result of the second set of tests. It can be observed that for the 11 test users, the accuracy, sensitivity and specificity of the system to detect no presence in bed is 100%.User sat on the bed edge and motion detected: This second case is different, as it combines information from two different sources. On the one hand, it is necessary that the presence detection service detects that the user is sitting on a bed edge. Table 6 shows the metrics for the positions sitting right and sitting left, which are related to knowing when the user is sat on the bed. To reduce the metrics of these positions to only one position, the arithmetic mean can be used, resulting in an accuracy of 96.21% ((100+92.42)/2), specificity of 100% ((100+100)/2) and a sensitivity of 77.27% ((100+54.54)/2). On the other hand, the motion detector needs to determine that the user is standing up when, according to the presence detector, the user is sitting down. The metrics in Table 7 when the user goes from lying down to sitting down are 96.25% accuracy, 100% specificity and 92.5% sensitivity. Considering the metrics of both cases, the metrics of this case can be estimated by treating it as an intersection of probabilities of independent events (P(A∩B)=P(A)P(B)), and therefore the estimated metrics would be
Accuracy: 0.9621 × 0.9625 × 100 = 96.60%Specifiicty: 0.1 × 0.1 × 100 = 100%Sensitivity: 0.9250 × 0.7727 × 100 = 71.47%According to the results, the system has good accuracy, and, with a high specificity, it is able to detect all TN correctly. The main problem of the system is to detect TP, having a low sensitivity. In this case, accuracy results in a misleading metric again. Obviously, this is an estimation, and the real metrics of the system could change.

Given these data, it is possible to say that there is a 100% probability of detecting when a user exits the bed, and it is estimated that it is possible to detect when a user is sitting on bed and ready to exit the bed with an accuracy of 96.60%. Analyzing the other metrics, the system does not raise false alarms, thanks to a great specificity of 100%, but also it is not able to detect all the situations where the user is ready to exit the bed, with a low sensitivity of 71.47%.

## 6. Discussion

This paper presents a system for both fall prevention and fall detection. To this end, the approach presented here differs from most of the state-of-the-art works in that the two topics (i.e., fall prevention and fall detection) are addressed simultaneously. Furthermore, these two topics are addressed in a very specific context, as it is bedtime for older adults. It is during bedtime when most falls take place. For this reason, this work focused on this specific context.

The proposed system is therefore aimed toward a twofold purpose. On the one hand, the system is intended to prevent falls from occurring and, on the other hand, to detect when a fall has occurred. From the fall prevention perspective, the proposed approach is based on identifying when a bed exit is taking place. This action, on its own, is considered a situation of risk for certain individuals. The proposed approach achieved promising results, combining information gathered from the IMU and the pressure sensors located underneath the mattress. The system reliably detects when a user exits the bed with an accuracy of 100%. When this situation is detected, the caregiver is notified, in real time, so that the older adult can be attended, thus contributing to ensure the safety of the older adult.

Additionally, this approach for fall prevention is also employed to detect the individuals’ intention to exit the bed, obtaining an estimated accuracy of 74.37%. This functionality is very useful as, besides detecting 100% of the situations in which the individual has exited the bed, the simple fact of attempting to do so already implies a risk of falling. This functionality enriches the fall prevention system. It is therefore very relevant to be able to alert the caregiver as early as possible. In general, the literature on fall prevention is quite limited compared to the fall detection one, which is usually restricted to gait or activity analysis in order to detect risk situations. Although the works in the literature aim to cover a wider range of situations and times of day, compared to the work presented here, in the end, the results obtained are not very reliable, given that they either simply detect users who may be more likely to fall or they involve a significant loss of privacy, as many of these systems use computer vision. As opposed to these works, the system proposed here focuses on that time of the day when individuals are most vulnerable and when privacy concerns are more stringent. In view of the results, the system may be of great use for caregivers either at home or in a nursing home.

The approach for fall detection is based on the use of a non-invasive and comfortable sensor attached to the user waist. While placing the sensor in a wristband on the wrist may be a more natural location for the user, in general, those solutions that opt for this location are more fragile, as a sensor on the arm will generate much more noise than one located on the waist. Environmental sensors, which are less physically invasive (no sensors are required to be placed on the user body), are also less intrusive, but ultimately lead to an invasion of privacy, as these environmental sensors are typically based on image recording. Finally, another option that is becoming widespread is the use of smartphones, as these have accelerometers and gyroscopes that provide the necessary information for fall detection. These types of solutions are not realistic when the objective is to monitor users in their homes, as it is unreasonable to expect the user to carry their mobile phones all over the house at all times.

Regarding the coverage range of the proposed solution, it is limited to about 5–10 m within the range of the processing node that downloads and classifies the data collected by the IMU, which may mean that the user can only be monitored in a single room. However, this problem can be solved by adding more processing nodes given the proposed system architecture. The IMU publishes events whenever it detects a possible fall event via BLE. When the node receives that event, it downloads the data and classifies the movement as a fall or an ADL. Scaling the range just requires additional Raspberry Pi nodes to be installed over the house that run the fall detection service and listen for IMU events. Note that all IMU-based solutions in the literature suffer from this limitation, as the BLE connection is generally a short-range one. Solutions based on cameras and environmental sensors also suffer from similar problems as they are static sensors and would have to be replicated with the cost that this entails. The smartphone solution, which is not affected by the coverage range problem, is not a viable solution for home environments as stated above.

It is worth noting that, unlike a large number of works in the literature, this article not only proposes a solution for fall detection, but also proposes an architecture capable of operating and detecting falls in real time, showing reliable results with an accuracy of 93.51%, sensitivity of 92.04% and specificity of 95.45%.

The achieved sensitivity indicates the rate of true positive being correctly identified. In this context in which the detection of a fall launches a notification to the caregiver, who has immediately attend the fallen, the need to avoid false alarms is very important. If at some point the caregiver has the feeling that the system commonly notifies false positive alerts, it is also possible that notifications are not given the appropriate importance. Specificity measures, on the other hand, the number of true negatives correctly identified. In this sense, it is also very important that events classified as non-falls are not, indeed, a fall. Having a low specificity will result in the system missing fall events and, in this situation, fallen individuals would be unattended until discovered by other means.

The comparison with other IMU-based systems from the literature (Table 8) shows that there are many proposals with better performance, many of them using neural networks. The use of this technique is proposed as future work to improve system performance, in addition to the collection of more fall data.

Despite the promising results obtained by the proposed approaches, based on the obtained accuracy rates, there are some limitations that need to be addressed in order to improve the system. The fall detection analysis is performed on the gateway because the required resources for the processing exceed the capability of the sensor. The use of the gateway is necessary if caregivers are to be notified when a relevant event takes place (i.e.,: bed exit or detected fall). Nonetheless, following the edge computing paradigm, ideally, the processing should remain as close to the information source and destination as possible. Efforts are currently being addressed to identify a sensor with support for running machine learning models, whereas at the same time, the proposed algorithm needs to provide a model, light enough, to support real-time processing in the sensor. Those approaches based on smart phones do not need to deal with this limitation, as current phones count on both the resources and connectivity to process locally and publish notifications when needed.

## 7. Conclusions and Future Work

This research works on the hypothesis that to achieve a system for fall prevention and fall detection that can be widely adopted by both older adults and caregivers, the system has to address the following requirements: being privacy aware, working in real time, avoiding the need to carry uncomfortable devices, and having a constrained price. Furthermore, the system performance has to minimize the false negative rate and achieve a similar performance to the best identified previous works. To validate this hypothesis, this work presents a system architecture for monitoring older adults, with some degree of dependency, during bedtime. The proposed system is intended to detect when the user is about to exit the bed so that caregivers can be notified as soon as possible to provide assistance and supervise the situation. The proposed system achieved a solution that ensures a low-cost solution, as it only requires a Raspberry Pi 4, the IMU Puck.js sensor and three pressure sensors. Moreover, the installation is very simple, and it ensures user privacy in contrast to other solutions based on video. Finally, the wearable sensors are of a limited size to be placed on the waist, thus being comfortable for the user and unnoticeable by others.

The system was experimentally validated by running a set of experiments in which the targeted activities were simulated, such as falling down or exiting the bed. The obtained results demonstrate the reliability of the system, achieving an accuracy of 100% for detecting the risk of fall, and an accuracy of 93.51% for detecting actual falls.

Future work will focus on the fall detection and prevention system in parallel. Regarding the fall detection system, improvement will be addressed to increase the accuracy rate, as it is essential to increase sensitivity. One option for improving accuracy rates involves implementing a neural network for the classification algorithm and collecting more fall data to improve the model. Another limitation of the current approach is the short range of the system, which limits the monitoring range to a single room. Although this problem can be addressed by using more hardware nodes, it is not ideal, so there are two alternatives that can be considered. The first option would be to create a sensor with a WiFi module that can send data to be processed in the cloud from anywhere in the home (nowadays, WiFi connections can be found in any corner of a house). However, this would have to be further studied, as WiFi modules consume much energy. The second option would also involve creating a node, which in this case would be able to carry out the processing at the edge, communicating only the results to a gateway to notify the caregiver. This gateway could be a mobile phone for example, and could extend the range of operation even outside the home.

Future work for the fall prevention system will focus on improving bed position detection so that this can help in improving the 74.37% accuracy achieved in detecting the intention to exit the bed. This improvement will undergo a second round of testing to see if the problem in the sitting right posture detection results was indeed due to a problem in one of the sensors, after which more sophisticated techniques for bed position classification will be searched. Moreover, the capability to monitor bed positions opens the door to other functionalities. In nursing homes, for example, there are many residents who require postural changes by a caregiver throughout the night. This system will enable the automatic tracking of the period of time that the person has been in a given position. This can help caregivers to record these changes to track that the person has been changed in their position in a timely manner to avoid the consequences of being immobilized in bed. This sort of monitoring could be performed without any invasion of privacy, as only the signals received by the pressure sensors would be processed. Furthermore, the monitoring of individual movements in bed could also be extended to measure the quality of sleep, as this depends on the movements made by the user throughout the night.

## Figures and Tables

**Figure 1 ijerph-19-07139-f001:**
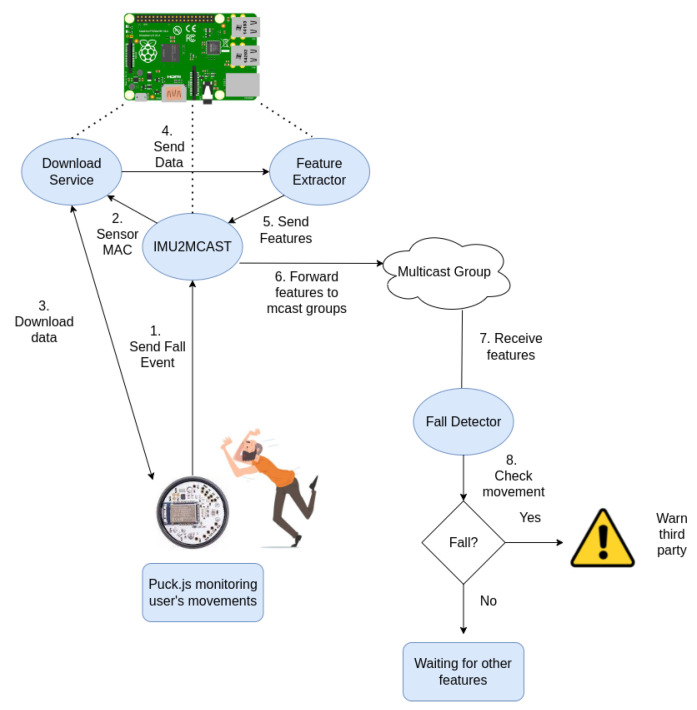
Proposed system architecture.

**Figure 2 ijerph-19-07139-f002:**
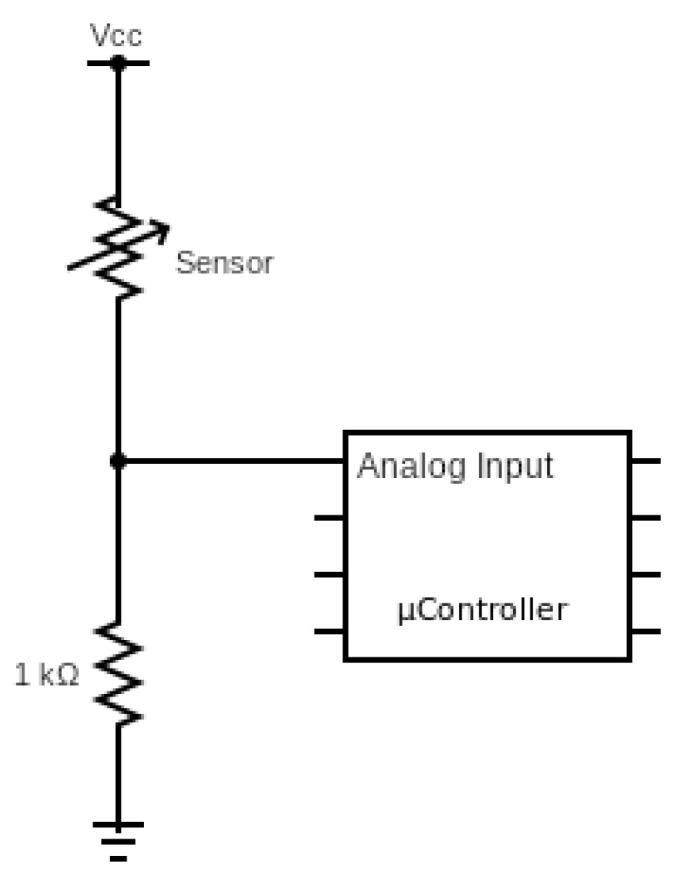
Connection circuit for analog sensors.

**Figure 3 ijerph-19-07139-f003:**
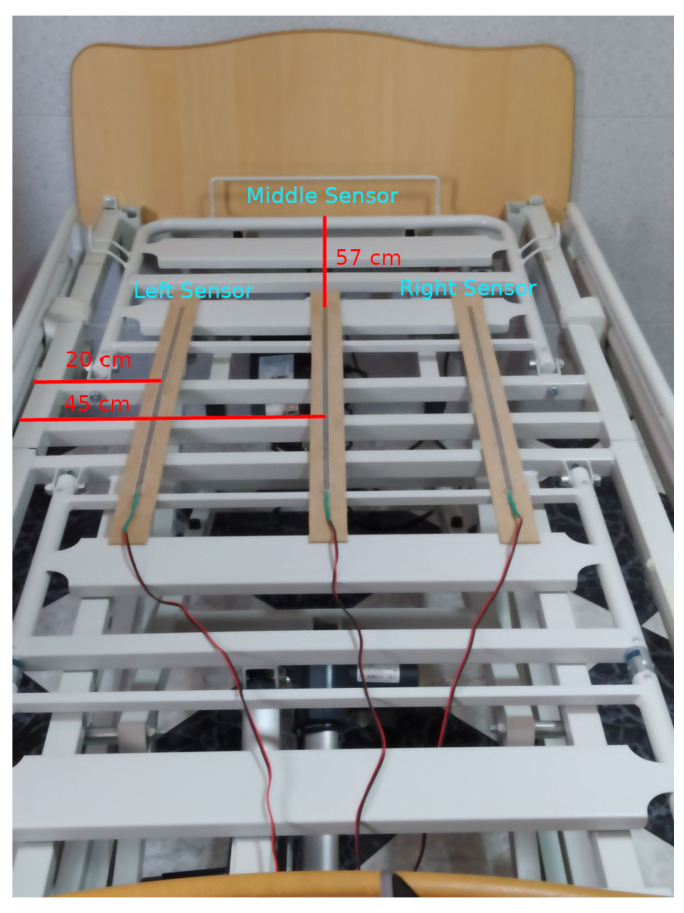
Location of sensors under the bed.

**Figure 4 ijerph-19-07139-f004:**
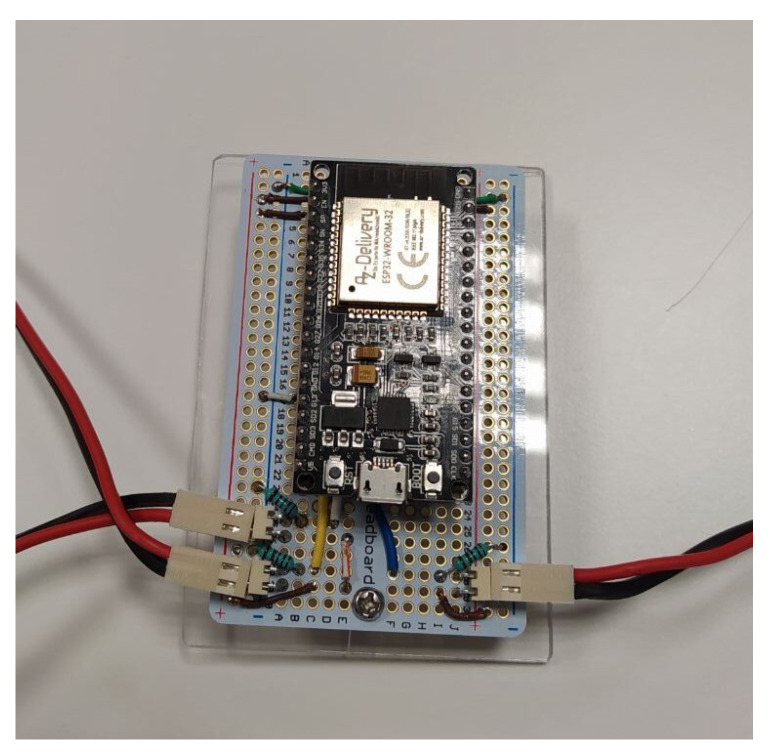
Processing node used for in-bed monitoring.

**Figure 5 ijerph-19-07139-f005:**
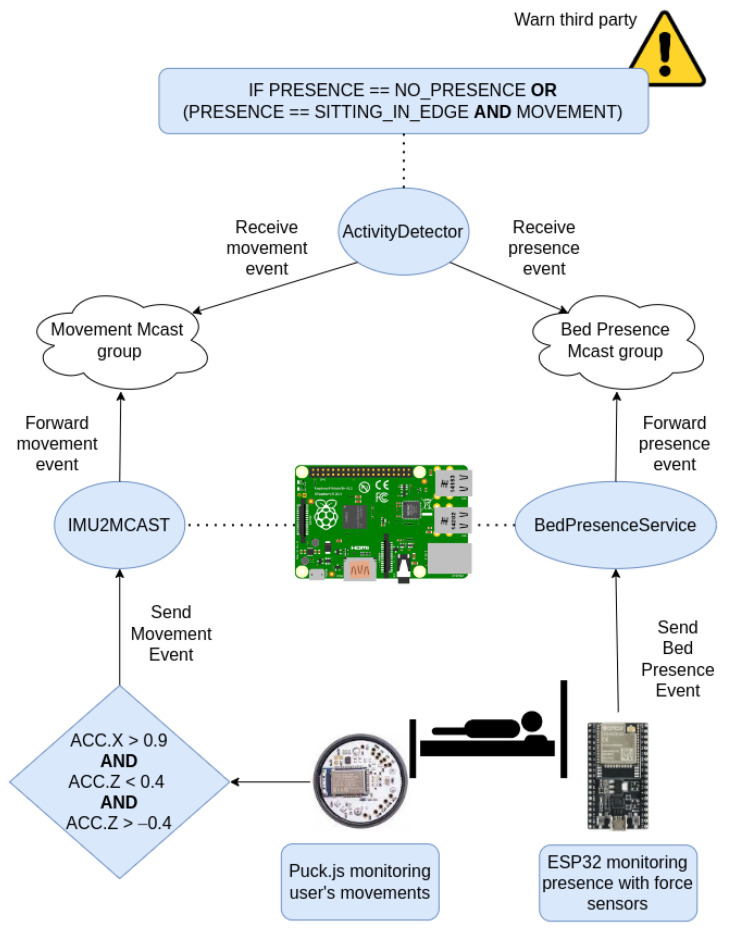
Overview of the fall-prevention system.

**Figure 6 ijerph-19-07139-f006:**
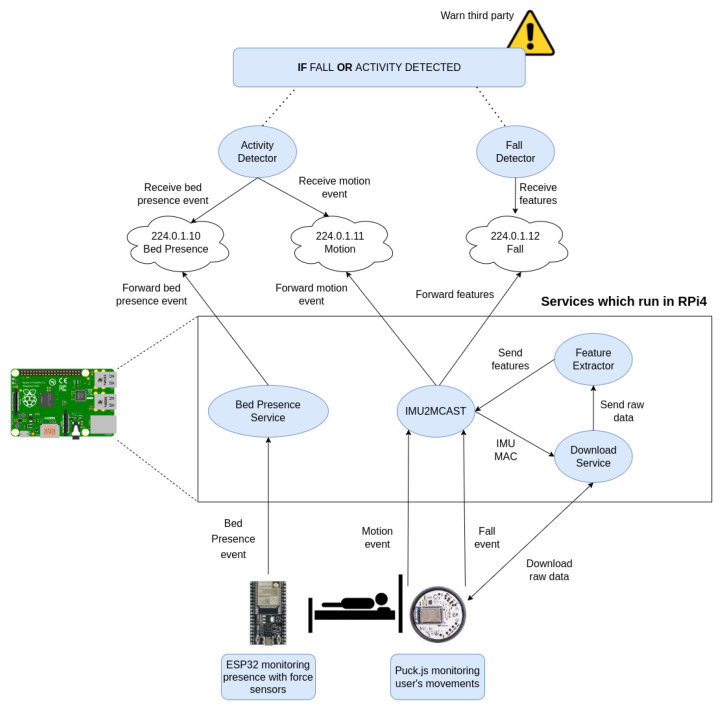
Diagram showing how the system works in a general way.

**Figure 7 ijerph-19-07139-f007:**
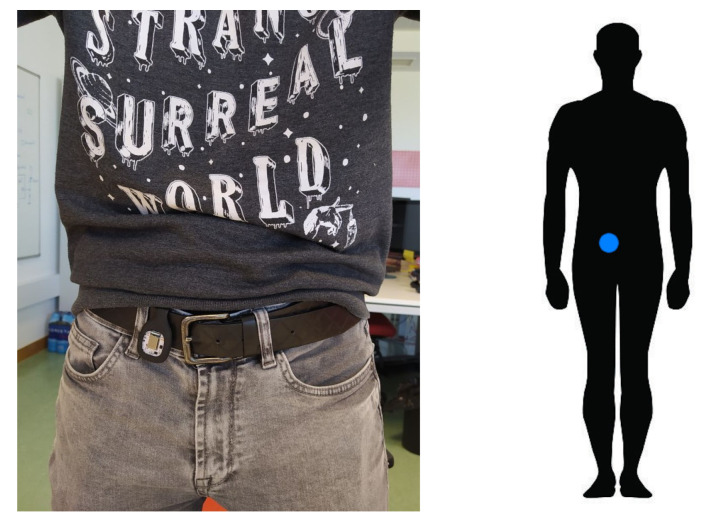
Location of the sensor in the user’s body.

**Figure 8 ijerph-19-07139-f008:**
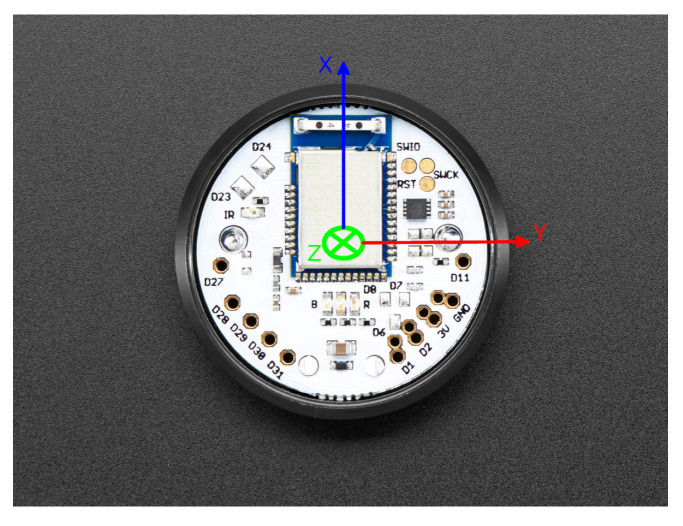
IMU sensor used for this work with its respective axis.

**Figure 9 ijerph-19-07139-f009:**
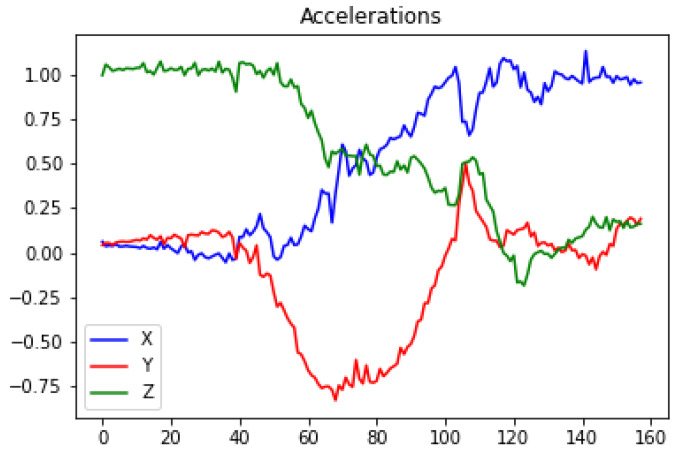
Accelerations of the users getting out of bed from left.

**Figure 10 ijerph-19-07139-f010:**
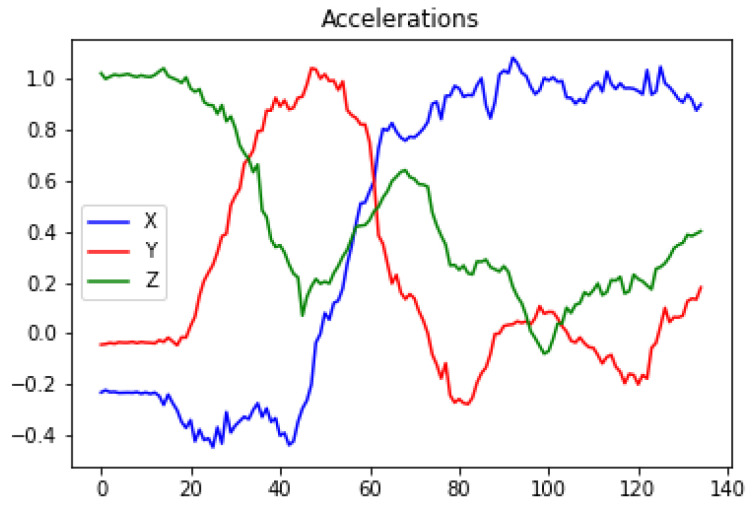
Accelerations of the users getting out of bed from right.

**Figure 11 ijerph-19-07139-f011:**
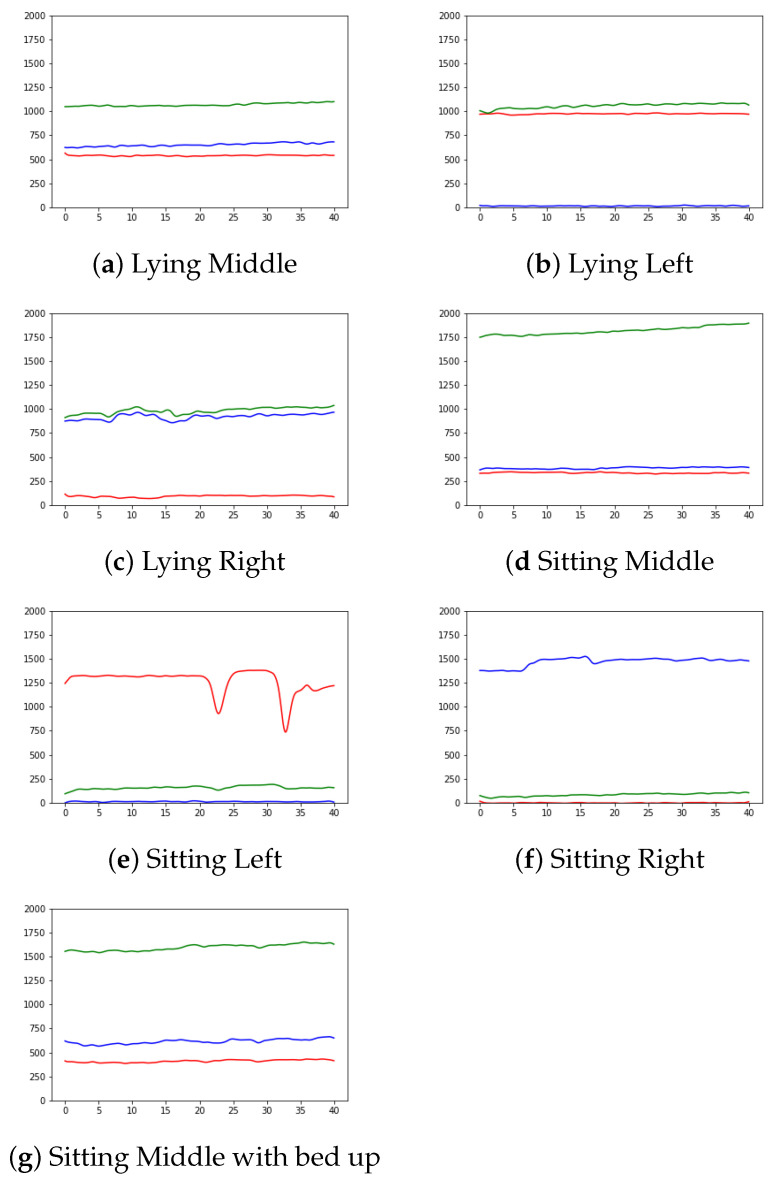
Sensor value depiction according to the position of the user on the bed. Each color is associated with a sensor, red with the left sensor, green with the central sensor and blue with the right sensor.

**Table 1 ijerph-19-07139-t001:** Summary of data collection methods for fall detection.

Wearable	Smart Bands	[7,8]
	Clothing	[9,10,11,14]
	Smart Phone	[18,19,21]
Ambient Sensors	Doppler	[23,24]
	UWB	[25]
	Infrared	[26]
	WiFi	[27,28]
Vision	Depth Camera	[29,30,31,32]
	RGB Camera	[33,34,35,36,37]

**Table 2 ijerph-19-07139-t002:** Table with the results of the tests grouped by exercise.

Activity	TP	TN	FP	FN
Backward Fall	17			5
Forwad fall	22			
Left fall	21			1
Right fall	21			1
Run		20	2	
Jump		22		
Sitting Down		21	1	

**Table 3 ijerph-19-07139-t003:** Results for the first battery of tests.

Movement in Bed	Type	Starting Position	Intermediate Positions	End Position
Lying middle to sitting right edge	Result Expected	Lying Middle	Lying Right	Sitting Right
	Result Achieved	Lying Middle	Lying Right	Sitting Right
Lying middle to sitting left edge	Result Expected	Lying Middle	Lying Left	Sitting Left
	Result Achieved	Lying Middle	Lying Left	Sitting Left
Lying middle to lying left border	Result Expected	Lying Middle	-	Lying Left
	Result Achieved	Lying Middle	-	Lying Left
Lying left to lying middle	Result Expected	Lying Left	-	Lying Middle
	Result Achieved	Lying Middle	-	Lying Middle
Lying middle to lying right	Result Expected	Lying Middle	-	Lying Right
	Result Achieved	Lying Middle	-	Lying Right
Lying right to lying middle	Result Expected	Lying Right	-	Lying Middle
	Result Achieved	Lying Right	-	Lying Middle
Getting out of bed from the left side	Result Expected	Lying Middle	Lying Left-Sitting Left	No presence
	Result Achieved	Lying Middle	Lying Left-Sitting Left	No presence
Lying on the bed from the left side	Result Expected	No presence	Sitting Left-Lying Left	Lying Middle
	Result Achieved	No presence	Sitting Left	Lying Middle
Getting out of bed from the right side	Result Expected	Lying Middle	Lying Right-Sitting Right	No presence
	Result Achieved	Lying Middle	Lying Right-Sitting Right	No presence
Lying on the bed from the right side	Result Expected	No presence	Sitting Right-Lying Right	Lying Middle
	Result Achieved	No presence	Sitting Right	Lying Right

**Table 4 ijerph-19-07139-t004:** Results for a real end user from the El Salvador nursing home.

Movement in Bed	Type	Starting Position	Intermediate Positions	End Position
Exit bed from the right side	Result Expected	Lying Middle	Lying Right-Sitting Right	No Presence
	Result Achieved	Lying Middle	Lying Right-Sitting Right	No Presence
Lying on the bed from the right side	Result Expected	No presence	Sitting Right-Lying Right	Lying Middle
	Result Achieved	No presence	Sitting Right-Lying Right	Lying Middle

**Table 5 ijerph-19-07139-t005:** Confusion matrix with the results of the second battery of tests.

Actual Class							
		No Presence	Sitting Right	Lying Middle	Lying Right	Lying Left	Sitting Left
**Predicted Class**	No Presence	11	0	0	0	0	0
	Sitting Right	0	11	0	0	0	0
	Lying Middle	0	0	11	1	0	0
	Lying Right	0	0	0	10	0	0
	Lying Left	0	0	0	0	11	5
	Sitting Left	0	0	0	0	0	6

**Table 6 ijerph-19-07139-t006:** Metrics for the multi-class confusion matrix.

Metric	No Presence	Sitting Right	Lying Middle	Lying Right	Lying Left	Sitting Left	System ($\bar{X}$)
Accuracy	100%	100%	98.48%	98.48%	92.42%	92.42%	96.97%
Sensitivity	100%	100%	100%	90.09%	100%	54.54%	90.91%
Specificity	100%	100%	98.18%	100%	90.91%	100%	98.18%

**Table 7 ijerph-19-07139-t007:** Results of movements test with IMU.

Phase	Accuracy	Specificity	Sensitivity
Lying-Sitting	96.25%	100%	92.5%
Sitting-Standing	100%	100%	100%
Standing-Walking	100%	100%	100%

**Table 8 ijerph-19-07139-t008:** Comparison of different fall detectors with the one proposed in this work.

Work	Accuracy	Algorithm	Location
This work	93.51%	SVM	Waist
[56]	98.61%	FD-CNN	Waist
[19]	96.75%	CNN-LSTM	Pocket
[17]	99.38%	TBA	Pocket
[58]	99.30%	TBA	Pocket
[50]	99.06%	Deepnet	Waist

## Data Availability

Data used for training (fall-dataset) and testing (fall-dataset2) can be found at https://arcoresearch.com/2021/04/16/dataset-for-fall-detection/ (accessed on 5 April 2022).

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
