# Peer review of "Bedtime Monitoring for Fall Detection and Prevention in Older Adults"

_ijerph, 2022, doi:10.3390/ijerph19127139_

Round 1
Reviewer 1 Report
I found the article very well structured and written. The scientific content is sound and a quite exhaustive state of the art has been conducted.
My only suggestion is to correct the following typos:
8: designed FOR bedtime
15: results show
232: individual on the BED
233: is that IT works
238: at WITH A low cost (at low cost)
275: exit BE is detected
296: remove advanced equipped
337: the proposeD
338: I'd rather say: the proposed method relies on a trained model
350: but IT stillS let them
354: set of testS
359: data WERE captured
360: data WERE used
393: publishES
396: downloadS
400: yieldED
421: or on A separate one
424: the one LISTENING
425: the data ARE downloaded
426: This data ARE
479: to obtain--
630: biase-
709: only two of the ARE erroneous
718: show-
759: on young persons
Best regards.
Reviewer 2 Report
I would like to thank the authors for their contribution. My recommendation for the manuscript will be minor revision. Please find my general opinion and some specific notes below.
Having a family member who worked at an elderly care home, I am a little skeptical of the usefulness of wearable sensors. But you have also proposed a bed sensors approach which may compensate for the disadvantages of the wearable sensor approach. I think you have a very good understanding of the literature and the shortcomings of your strategies. Using ML/AI methods as suggested in your future work can help reduce reliance on the sensors, offering a best of both worlds so to speak. Considering an average elderly care home and how it works, the maintenance of these devices can become quite problematic as well because we expect hundreds of sensors for the entire building and it will demand great attention from the staff to make sure of their functioning.
Notes:
- Provide referencing within the text for the Table 1.
- Table 2 contains several figures which are badly oriented and missing labels.
- Some other figures are missing their labels as well.
- On page 16, line 570 correct "patter" as "pattern".
- As an addition to the Figure 1, it would be nice to show a picture of the IMU on an actual test subject from the experiments.
- If the online version of the paper can include a video of the fall detection/prevention, it would be useful for the readers to see. If the MDPI provides this, I highly recommend to upload one.
- Please provide formulas for the specificity and the sensitivity calculations, similar to the accuracy calculation formula described on Page 21 line 725.
Reviewer 3 Report
- Include the detail of performance metrics for more clarity and understanding .
- Comparison and analysis section require further enhancement.
- Add Pseudocode Examples of your proposed and based method.
- Please display some discussion about the results to explain the drawbacks of the proposed method.
- What is the main contributions of this article? In general, the article needs to further refine the scientific issues to be resolved. The current status of the research should clearly state what the current scientific problem is and what scientific problem the paper solves.
